# Transmission Parameter Design and Characteristic Analysis of Three-Row Parallel Planetary Gear HMCVT

**Huadong Zhou [1], Lin Wang [2], Zhixiong Lu [1,*], Jin Qian [1], Haijun Zhang [1], Yirong Zhao [2], Zhun Cheng [3] and Xingwei Wang [2]**

[1] College of Engineering, Nanjing Agricultural University, Nanjing 210031, China
[2] State Key Laboratory of Power System of Tractor, Luoyang 470139, China
[3] Department of Vehicle Engineering, College of Automobile and Traffic Engineering, Nanjing Forestry University, Nanjing 210037, China
* Correspondence: luzx@njau.edu.cn

**Abstract:** A three-planetary-row linkage HMCVT scheme is designed for the operation requirements of low speed and high torque of tractors. The HM1 section adopts hydraulic mechanical power to begin its operations so that the tractor can obtain a greater transmission ratio at a low speed. The HM2 and HM3 sections adopt an "equal ratio" design so that the system has a better speed-regulation performance. The clutches controlling forward and backward movements are placed on the output shaft so that the forward and backward sections have a wider speed range. The stepless speed-regulation characteristics and torque characteristics of HMCVT are analyzed, and they can meet the kinematic and dynamic requirements. The transmission ratios of the three sections are as follows: HM1 section, 14-1.85; HM2 section, 1.85-1.04; HM3 section, 1.04-0.6. The corresponding tractor speed ranges are as follows: HM1 section, 0.2–14 km/h; HM2 section, 14–26 km/h; HM3 section, 32–46 km/h. According to energy conservation, the transmission efficiency of the system is analyzed in combination with the power flow characteristics; the highest transmission efficiencies are as follows: HM1 section, 0.85; HM2 section, 0.88; HM3 section, 0.92. When the system has cycle power, the overall transmission efficiency of the system is low and is greatly affected by the change in displacement ratio; when the system does not have cycle power, the transmission efficiency is less affected by the displacement ratio.

**Keywords:** three-planetary-row linkage; low speed and high torque; HMCVT; characteristic analysis

## 1. Introduction

The working environment of tractors is complex and changeable, and it is necessary to frequently switch gears to meet the speed and power requirements [1]. Traditional tractors usually increase the number of gears to achieve higher target transmission ratios and meet the complex and changeable speed requirements [2,3]. The increase in the number of gears undoubtedly increases the complexity of the transmission system. The frequent switching of gears not only greatly reduces the work efficiency, but also makes it difficult for the driver to ensure comfortable operation [4–6]. The hydro-mechanical continuously variable transmission (HMCVT) is a new form of transmitting hydraulic power flow and mechanical power flow in parallel; it is composed of a hydraulic speed-control mechanism, mechanical transmission mechanism, and shunt and confluence mechanism. The transmission efficiency is realized through mechanical transmission, and the continuously variable transmission is realized through the combination of hydraulic transmission and mechanical transmission. The continuously variable transmission of hydraulic machinery can automatically adjust the output transmission ratio to adapt to the changing load and vehicle speed so that the transmission outputs with a continuous transmission ratio to ensure that the engine works at the best point [7–10].

The transmission scheme design is one of the key and difficult points in HMCVT technology [11,12]. A reasonable transmission scheme guarantees good output characteristics for the system [13]. According to the different dynamic coupling modes, the HMCVT scheme is divided into three types: "output coupling", "input coupling", and "hybrid". Depending on the number of planets, these are divided into "single planets" and "multiple planets". Xu et al. [14] designed a single-row, star-row multi-stage scheme, using a set of planetary gear mechanisms coupling hydraulic power and mechanical power output, by controlling the pump–motor system displacement changes to achieve a continuous output transmission ratio. Zhu et al. [15] conducted a comparative analysis of the single-planetary-row and double-planetary-row structures and designed a hydraulic speed-control system for the hydraulic, mechanical, stepless gearbox; the gearshift clutch control system of the gearbox; the engine power-matching system; and the power-matching system of the human–machine integration control to meet the transient dynamic performance requirements and steady-state economic requirements of the vehicle during operation. Yu et al. [16–18] designed a hybrid hydraulic mechanical, continuously variable transmission scheme, which is composed of two parallel groups of differential gear trains and a pump–motor system. The speed-regulation characteristics of the hydraulic mechanical composite transmission can be made more stable by connecting the two differential gear trains, which provides advantages over the output coupling scheme in terms of load capacity and speed-regulation characteristics.

For single-planetary and double-planetary structures, the transmission ratio that can be achieved is limited, and for tractors that often work at low speeds, a larger transmission ratio output is needed to meet their power requirements [19,20]. A three-planetary-row linkage HMCVT scheme is proposed to realize the low-speed and high-torque operation of tractors. The parallel structure of the three-planetary-row linkage confluence mechanism and the variable pump–quantitative motor hydraulic speed-control system are designed, and the transmission parameters of the designed scheme are matched according to the target speed ratio requirements. According to the actual steady-state output characteristics of the engine, the speed-regulation characteristics, power shunt characteristics, torque characteristics, and efficiency characteristics of the designed HMCVT scheme are analyzed.

## 2. Materials and Methods

### 2.1. Three-Planetary-Row Manifold Scheme Design

To allow for the low-speed and high-torque operation of tractors, a three-planetary-row linkage confluence structure is proposed. The hydraulic mechanical power is used to start, and the starting range of the transmission ratio is expanded. The planetary gear mechanism is made empty on the PTO axis using the empty sleeve structure so that the spatial layout is compact and reasonable. The clutches controlling forward and backward movements are placed at the power output end so that the backward and forward sections of the tractor have good speed-regulation characteristics. The schematic diagram of the design scheme is shown in Figure 1.

The designed HMCVT system consists of three forward sections, HM1, HM2, and HM3, and three backward sections, RHM1, RHM2, and RHM3, which are switched by different clutch control sections. The clutches' working states in different working sections are shown in Table 1.

The $C_1$, $C_2$, and $C_3$ clutches correspond to the switching control of the three working sections, respectively, while the CV and CR clutches control the forward and backward movement sections, respectively. HM1's power is output by $P_3$ gear ring $r_3$, HM2's power is output by $P_2$ sun gear $s_2$, and HM3's power is output by $P_1$ planetary carrier $c_1$. The power transmission routes are shown in Figure 2.

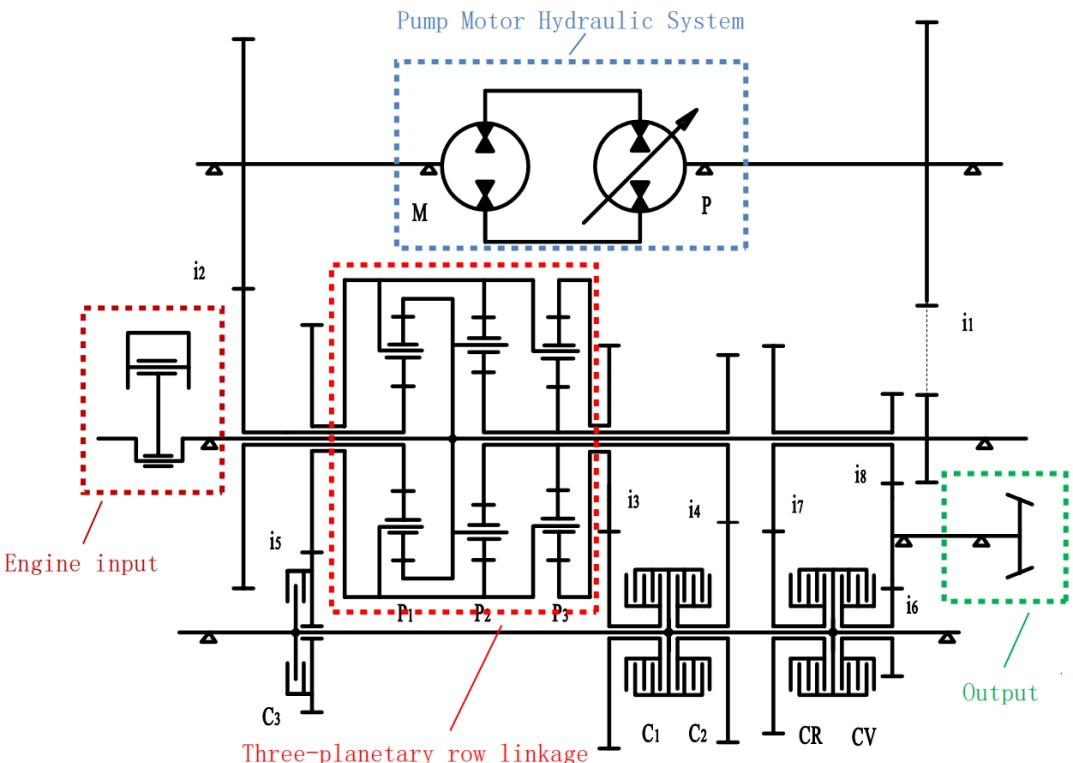

**Figure 1.** Principle diagram of three-planetary-row HMCVT scheme.

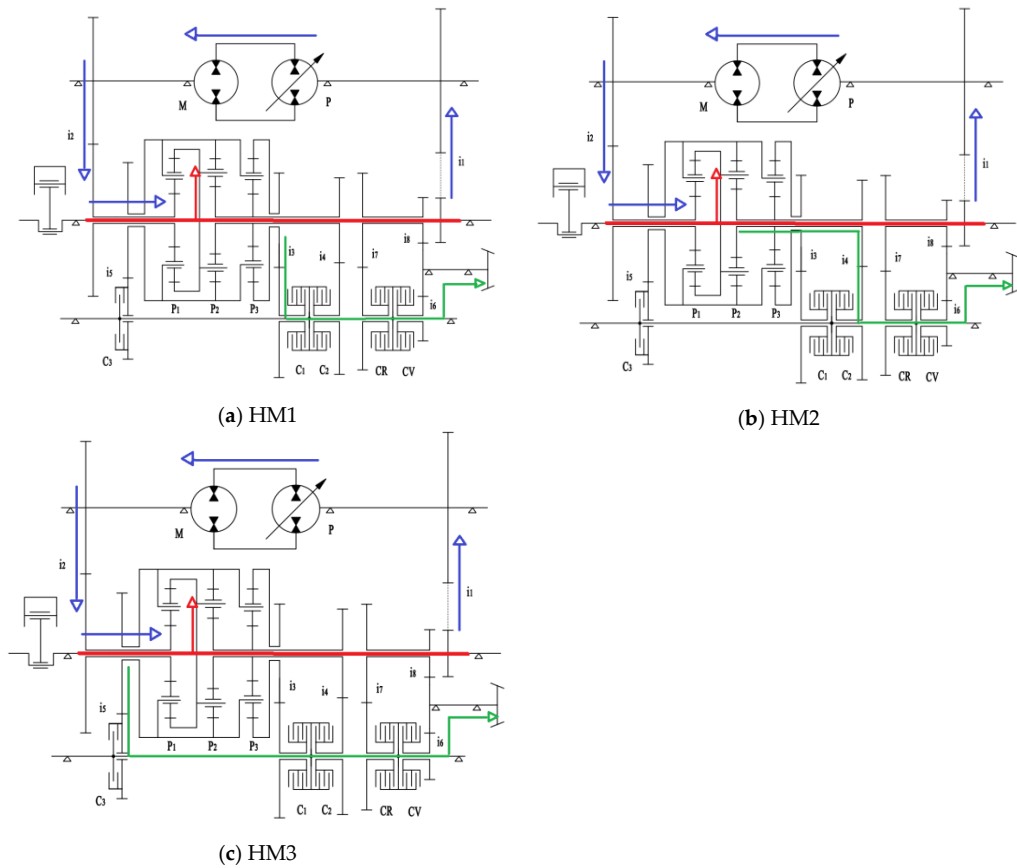

(**a**) HM1

(**b**) HM2

(**c**) HM3

**Figure 2.** Power transmission route of HMCVT forward sections.

**Table 1.** Working states of clutches in different working sections.

| Working Section | $C_1$ | $C_2$ | $C_3$ | CV | CR |
|:---:|:---:|:---:|:---:|:---:|:---:|
| HM1 | ● | ○ | ○ | ● | ○ |
| HM2 | ○ | ● | ○ | ● | ○ |
| HM3 | ○ | ○ | ● | ● | ○ |
| RHM1 | ● | ○ | ○ | ○ | ● |
| RHM2 | ○ | ● | ○ | ○ | ● |
| RHM3 | ○ | ○ | ● | ○ | ● |

● indicates that the clutch is in a closed state; ○ indicates that the clutch is in an open state.

In Figure 2, the red line represents the power emitted by the engine, the blue line represents the power transmitted by the hydraulic system, and the green line represents the power output after the confluence of the planetary gear mechanism.

*2.2. Design of HMCVT Transmission Parameters*

The target speed range of the design scheme is 0.2–40 km/h, the power radius of the tractor drive wheel is $r_d = 0.9$, the rear axle transmission ratio is $i_q = 0.38$, the wheel drive ratio is $i_b = 7.07$, and the engine rated speed is $n_e = 2200$ r/min. The ratio of the maximum output speed and the minimum output speed of each hydraulic mechanical section of the "equal ratio" transmission is constant, and the maximum output power of the system is constant. The torque gradually decreases with the increase in vehicle speed, the output torque is larger at low speeds and smaller at high speeds, which is suitable for tractor driving. Combined with the transmission characteristics of "equal ratio", to achieve the low-speed and high-torque operating characteristics, this paper adopts the idea of "class equal ratio" to design the starting HM1 section as the hydraulic mechanical power-starting; the HM2 and HM3 sections adopt the "equal ratio" design to ensure that the output transmission ratio is continuous, that the range of change is in the target value range, and that the transmission parameters are designed and matched. The main parameters are planetary row characteristic parameters $k_1$, $k_2$, and $k_3$ and fixed-axis-gear transmission ratios $i_1$, $i_2$, $i_3$, $i_4$, $i_5$, $i_6$, $i_7$, and $i_8$.

The rotational speed of each component of the planetary gear mechanism satisfies the relationship shown in Formula (1).

$$n_s + kn_r = (1+k)n_c \tag{1}$$

where $n_s$ is the sun wheel speed, $n_r$ is the gear ring speed, $n_c$ is the planet carrier speed, and $k$ is the planetary row characteristic parameter.

The output transmission ratio of HMCVT is equal to the ratio of engine speed to output shaft speed. The expressions of the output transmission ratio $I_{HM1}$, $I_{HM2}$, and $I_{HM3}$ and the output speed of the forward three sections $n_{HM1}$, $n_{HM2}$, and $n_{HM3}$ were obtained according to the analysis of the power transmission circuit diagram in Figure 2, as shown in Formulas (2)–(7).

$$I_{HM1} = \frac{(1+k_1)k_3 i_3 i_6}{(1+k_2+k_3)(\frac{e}{i_1 i_2}+k_1)-(1+k_1)(1+k_2)} \tag{2}$$

$$I_{HM2} = \frac{(1+k_1)i_4 i_6}{(1+k_1)(1+k_2)-k_2(\frac{e}{i_1 i_2}+k_1)} \tag{3}$$

$$I_{HM3} = \frac{(1+k_1)i_5 i_6}{\frac{e}{i_1 i_2}+k_1} \tag{4}$$

$$n_{HM1} = n_e \frac{(1 + k_2 + k_3)(\frac{e}{i_1 i_2} + k_1) - (1 + k_1)(1 + k_2)}{(1 + k_1)k_3 i_3 i_6} \tag{5}$$

$$n_{HM2} = n_e \frac{(1 + k_1)(1 + k_2) - k_2(\frac{e}{i_1 i_2} + k_1)}{(1 + k_1)i_4 i_6} \tag{6}$$

$$n_{HM3} = n_e \frac{\frac{e}{i_1 i_2} + k_1}{(1 + k_1)i_5 i_6} \tag{7}$$

where $e$ is the displacement ratio of the variable pump.

Reference [21] shows the tractor operating speed range in different modes. The target velocity range of the three working sections is designed to be HM1, 0.2–12 km/h; HM2, 12–24 km/h; HM3, 24–48 km/h. The ratio $\phi$ is 2.

The total transmission ratio range of HMCVT is determined by the target speed range, rear axle transmission ratio, wheel side transmission ratio, and wheel radius. The tractor speed and transmission ratio meet the relationship shown in Formula (8).

$$v = 0.377 \frac{n_e r_d}{i} = 0.377 \frac{n_e r_d}{i_q i_b i_{HM}} \tag{8}$$

where $i_{HM}$ is the gearbox output transmission ratio.

After substituting the minimum value 0.2 and maximum value 48 of the designed target vehicle speed into Formula (8), the range of the HMCVT output transmission ratio is 0.69–138.92. The HM1 section has the lowest vehicle speed, and the corresponding transmission ratio is the largest. When the displacement ratio $e = -1$, the transmission ratio $i_{HM1}$ reaches the maximum value of 138.92. Combined with Formula (2), the relationship shown in Formula (9) is obtained.

$$I_{HM1(e = -1)} = \frac{(1 + k_1)k_3 i_3 i_6}{(1 + k_2 + k_3)(-\frac{1}{i_1 i_2}) - (1 + k_1)(1 + k_2)} = 138.92 \tag{9}$$

The speed of the HM3 section is the highest, and the corresponding transmission ratio is the smallest. When the displacement ratio $e = +1$, the transmission ratio $i_{HM3}$ reaches the minimum value of 0.69. Combined with Formula (4), the relationship shown in Formula (10) is obtained.

$$I_{HM3(e = +1)} = \frac{(1 + k_1)i_5 i_6}{\frac{1}{i_5 i_6} + k_1} = 0.69 \tag{10}$$

According to the design requirements of "equal ratio" transmission, the ratio of maximum output speed to the minimum output speed of the HM2 and HM3 sections is equal. The output speed of HM2 is at its maximum when the displacement ratio $e = -1$ and at its minimum when the displacement ratio $e = +1$. The output speed of HM3 is at its maximum when the displacement ratio $e = +1$ and minimum when the displacement ratio $e = -1$. Combined with the speed output characteristics of Formulas (6) and (7), the relationship shown in Formula (11) is obtained.

$$\frac{(1 + k_1)(1 + k_2) - k_2(k_1 - \frac{1}{i_1 i_2})}{(1 + k_1)(1 + k_2) - k_2(k_1 + \frac{1}{i_1 i_2})} = \frac{k_1 + \frac{1}{i_1 i_2}}{k_1 - \frac{1}{i_1 i_2}} = 3 \tag{11}$$

From the continuous characteristics of the output transmission ratio, it can be seen that the minimum transmission ratio of the HM1 section is equal to the maximum transmission ratio of the HM2 section, and the minimum transmission ratio of the HM2 section is equal to the maximum transmission ratio of the HM3 section. The HM1 section has the smallest output transmission ratio when the displacement ratio $e = +1$; the output transmission ratio of the HM2 section is the largest when the displacement ratio $e = +1$ and the smallest when the displacement ratio $e = -1$. The output transmission ratio of the HM3 section is

maximum when the displacement ratio $e = -1$. The combination (2)–(4) obtains the relation shown in Formulas (12) and (13).

$$\frac{(1+k_1)k_3i_3i_6}{(1+k_2+k_3)(k_1+\frac{1}{i_1i_2})-(1+k_1)(1+k_2)} = \frac{(1+k_1)i_4i_6}{(1+k_1)(1+k_2)-k_2(k_1+\frac{1}{i_1i_2})} \quad (12)$$

$$\frac{(1+k_1)i_4i_6}{(1+k_1)(1+k_2)-k_2(k_1-\frac{1}{i_1i_2})} = \frac{(1+k_1)i_5i_6}{k_1-\frac{1}{i_1i_2}} \quad (13)$$

Reference [22] shows that the value range of the planetary gear characteristic parameters is generally $1.5 \le k \le 3$, and the value range of the fixed-axis-gear transmission ratio is generally $0.5 \le i \le 2$. In the three sections, the $P_1$ planetary row is involved in the work; larger characteristic parameters are more conducive to the gearbox achieving a large transmission ratio output, so $k_1 = 3$. HM3 is the over-speed section; the smaller the value of $i_5$, the better the adjustment of other gear transmission parameters, so $i_5 = 0.5$. The engine power is transmitted to the variable pump through gear $i_1$; in order to give full play to the regulation characteristics of the variable pump, $i_1$ is determined by the rated speed of the two. $i_7$ and $i_8$ are power transmission gears in backward working sections; their transmission ratio satisfies the relation $i_7 \times i_8 = i_6$ [23].

Combining Formulas (8)–(13), the characteristic parameters of planetary gears and the transmission ratio parameters of fixed-axis gears are shown in Table 2.

**Table 2.** Transmission parameters of three-planetary-row HMCVT scheme.

| $k_1$ | $k_2$ | $k_3$ | $i_1$ | $i_2$ | $i_3$ | $i_4$ | $i_5$ | $i_6$ | $i_7$ | $i_8$ |
|---|---|---|---|---|---|---|---|---|---|---|
| 3 | 2 | 3 | 0.78 | 1.25 | 1.5 | 1.6 | 0.5 | 1.2 | 1 | 1.2 |

In the process of parameter calculation and matching, the numerical value is adjusted appropriately considering the actual gear-matching problem, so the actual displacement ratio is not at $e = \pm 1$, but this has little effect on the speed-regulation performance of the HMCVT, which still meets the design requirements [24,25].

## 3. Results and Discussion

### 3.1. Results of Stepless Speed-Regulation Characteristics

The function of the HMCVT is to achieve a continuous change in the transmission ratio, and the HMCVT needs to have high transmission efficiency. Stepless speed-regulation characteristics, torque characteristics, power shunt characteristics, and efficiency characteristics are important indicators for the performance evaluation of an HMCVT, and these characteristics directly affect the performance of tractors [26,27].

Stepless speed-regulation characteristics mainly refer to HMCVT output transmission ratio characteristics and speed characteristics. The output transmission ratio must be able to continuously change in the target range value, while the tractor speed must be able to achieve the lowest and highest target values [28].

3.1.1. Transmission Ratio Characteristic Analysis Results

Because the clutches controlling forward and backward movements are placed on the power output shaft end, the output transmission ratio characteristics of the backward sections are the same as those of the forward section. Combined with the transmission parameter values in Table 2, according to the expression of the output transmission ratio of the forward section (2)–(4), the variation characteristics of the output transmission ratio of the designed HMCVT with the variable pump displacement ratio are obtained, as shown in Figure 3.

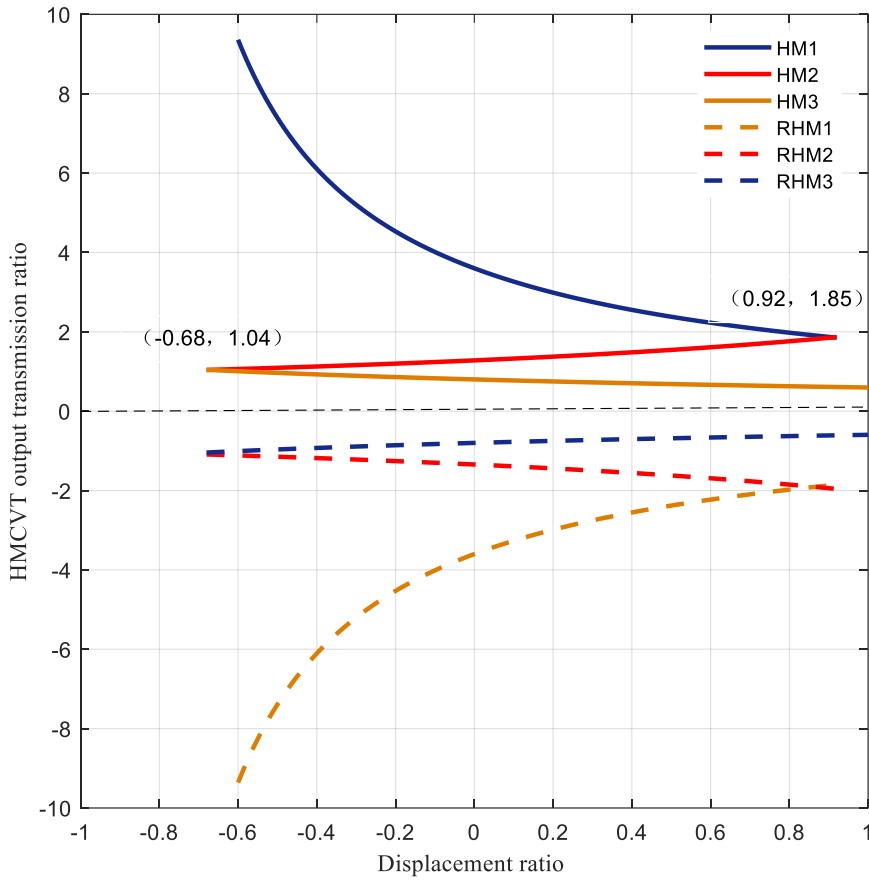

**Figure 3.** Transmission ratio output characteristic diagram of HMCVT.

Figure 3 shows that the output transmission ratio of the HMCVT system changes continuously. In the initial stage of HM1, the system starts with a large transmission ratio. With the change in displacement ratio from −1 to 0.92, the output transmission ratio gradually decreases to 1.85. In the HM2 section, the displacement ratio changes from 0.92 to −0.68, and the output transmission ratio gradually decreases to about 1. When the displacement ratio $e$ = 0.92, the HM1 section and the HM2 section can theoretically complete the synchronous section replacement. The HM3 section works alternately; the displacement ratio changes from −0.68 to 1, and the output transmission ratio gradually decreases to about 0.6. When the displacement ratio $e$ = −0.68, the HM2 section and the HM3 section can theoretically complete the synchronous section replacement. The output transmission ratio of the backward sections RHM1, RHM2, and RHM3 is symmetrical with the forward sections and has a good speed range. The forward and backward working sections are realized by switching the CV and CR clutches, and the continuous change in output transmission ratio is realized through the switching of $C_1$, $C_2$, and $C_3$ clutches. The three sections work continuously and alternately to meet the demand for continuously variable tractor transmission.

The transmission ratio output characteristics of the HM1 section are shown in Figure 4. When adjusting the variable pump displacement ratio to −0.9, the theoretical output transmission ratio of the HMCVT can reach 46. When the initial displacement ratio is adjusted, the HMCVT outputs a wide transmission ratio, allowing the tractor to achieve low-speed and high-torque operation.

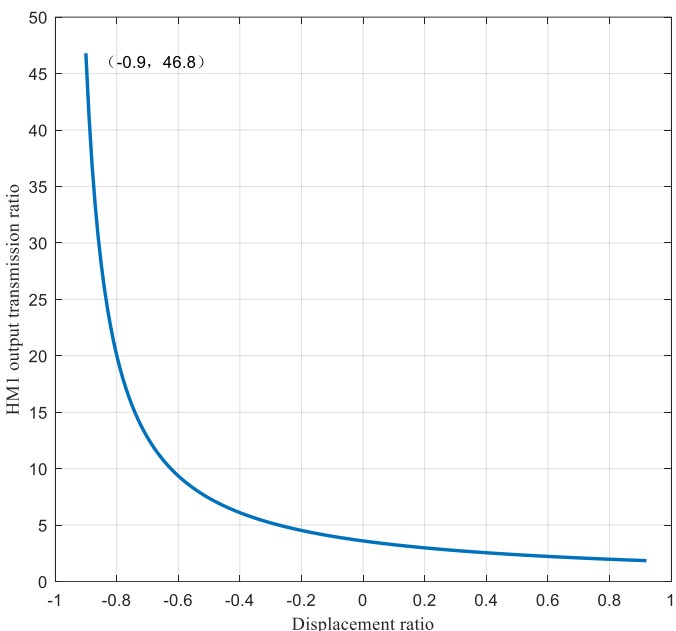

**Figure 4.** Transmission ratio output characteristics of HM1 section.

### 3.1.2. Speed Characteristic Results

The relationship between tractor speed and HMCVT output transmission ratio and engine output speed can be obtained using Formula (8). The engine adopts a constant speed-control strategy and sets engine output speed $n_e$ = 2500 r/min, 2200 r/min, and 1500 r/min. The relationship between the tractor speed and HMCVT transmission ratio is analyzed as shown in Figure 5.

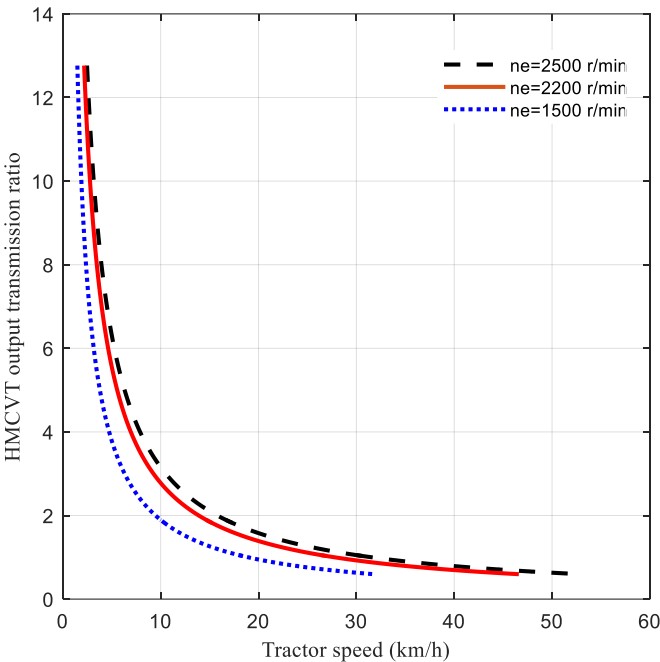

**Figure 5.** Relationship between HMCVT transmission ratio and vehicle speed at different engine speeds.

As shown in Figure 5, HMCVT can realize stepless speed-regulation at different engine speeds. When the engine speed is 2500 r/min, 2200 r/min, and 1500 r/min, the maximum speed of the tractor can reach 53 km/h, 46 km/h, and 32 km/h, respectively, which can

meet the basic operation requirements of tractors. Combined with the transmission ratio characteristics of Figure 3, the output transmission ratio of the HMCVT system can be increased by adjusting the value of the starting displacement ratio of the HM1 section to the −1 direction to reach the large transmission ratio required by tractors at a low speed.

### 3.2. Results of Torque Characteristics

The engine torque characteristics reflect the relationship between the output torque and the rotational speed of the engine under the condition of constant output power. Based on the engine test data, [29] uses the polynomial fitting method to construct the relationship between the steady-state output torque of the engine and the rotational speed and the throttle opening as follows:

$$T = 490 + 48\sin[\frac{pi}{2}(\frac{n_e}{800} - \frac{7}{8})] - \frac{10350}{\alpha(800 + 1680\alpha - n_e)} \tag{14}$$

where $pi$ = 3.14; $n_e$ is the engine speed, r/min; and $\alpha$ is the engine throttle opening, %.

The engine throttle opening is set to different opening states, $\alpha$ = 0.4, 0.6, 0.8, and 1, and the speed-regulation characteristics are shown in Figure 6.

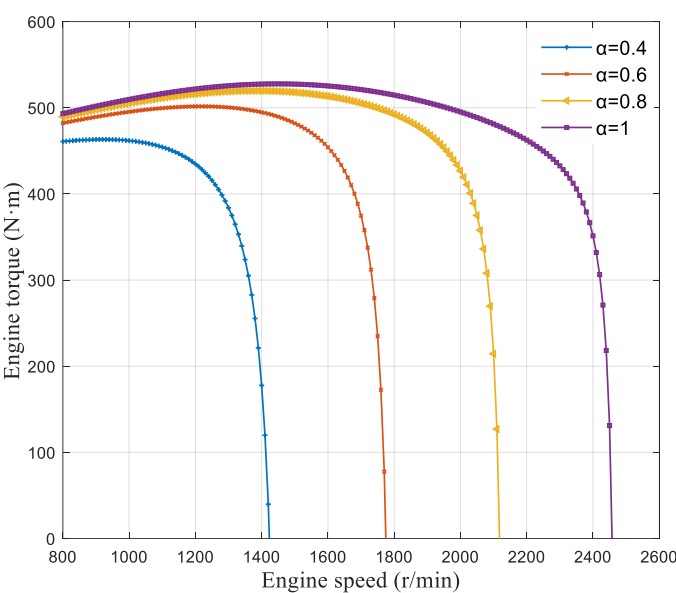

**Figure 6.** Engine speed-regulation characteristic diagram.

The torque characteristics of HMCVT refer to the ability to output the maximum torque after the torque change in the HMCVT system at different engine working stages, which is an important indicator of the dynamic performance of the HMCVT system [30,31]. By setting the engine throttle opening $\alpha$ = 1 and combining this with the speed-regulation characteristics of the engine in Figure 6, the relationship between the output torque of the HMCVT system and the change in the variable pump displacement ratio and the engine speed in the three forward working sections is obtained, as shown in Figure 7.

The results show that when the tractor is working in the low-speed section of HM1, the HMCVT can provide a wide transmission ratio and achieve a high torque output of up to 4000 N·m. When working at high speeds in HM3, the HMCVT output transmission ratio is less than 1, the system output torque is less than the engine output torque, and the maximum value is 550 N·m. In order to realize the maximum torque output of HMCVT, the engine should work at the best speed point. The analysis results of Figure 7 show that when the engine speed is 1600 r/min, the continuous output of maximum torque in HMCVT sections can be realized by adjusting the change in displacement ratio, which is consistent with the steady-state output torque characteristics of Figure 5.

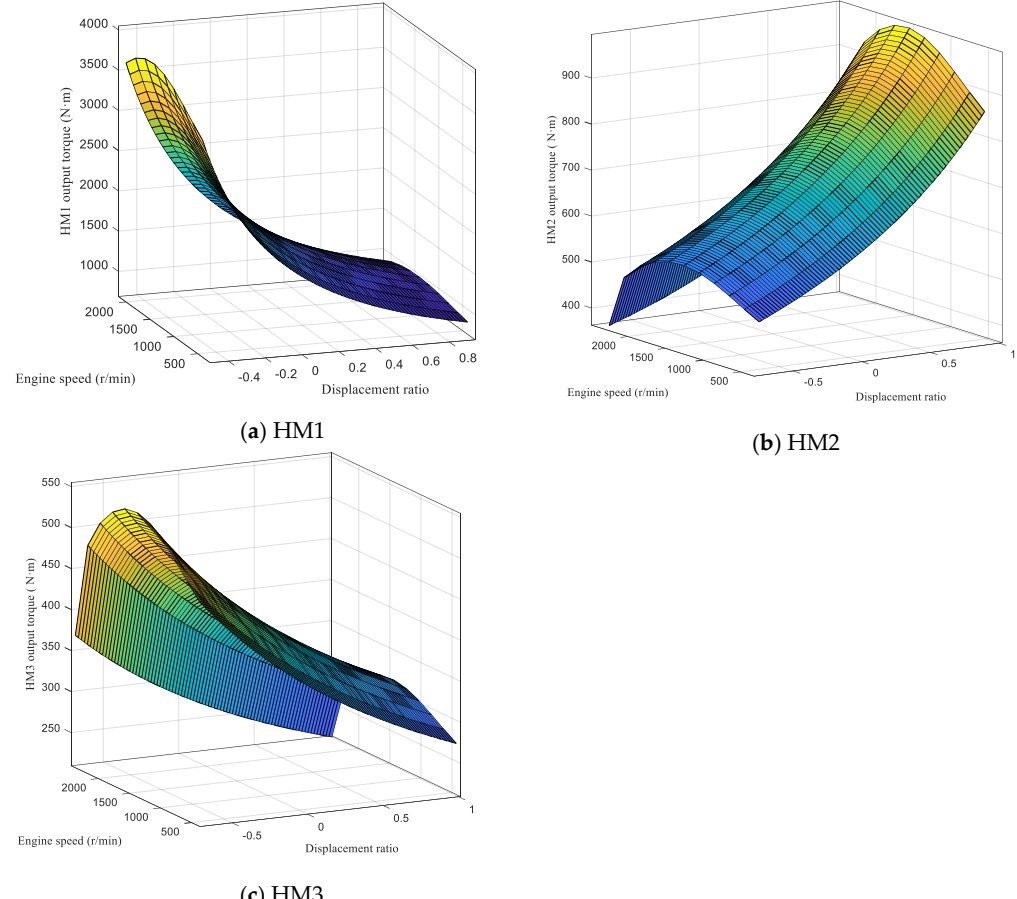

**Figure 7.** HMCVT torque characteristics.

### 3.3. Results of Power Shunt Characteristics

The power shunt characteristics of the HMCVT reflect the transmission path and direction of power in the HMCVT system. The power direction is determined by the direction of torque and speed. When the torque and speed at the node are in the same direction, the power is positive. When the torque and speed at the node are in opposite directions, the power is negative and the power flows from positive to negative [32,33]. Taking the HM1 section as an example, the power transmission route is shown in Figure 8: the red line is the mechanical power emitted by the engine, the blue line is the hydraulic power transmitted by the hydraulic system, and the green line is the power transmitted after the confluence of hydraulic power and mechanical power.

When there is cycle power in the closed hydraulic system, the power transmission route is as shown in Figure 8a. At this time, the mechanical power is input by the ring $r_1$ of the $P_1$ planetary row, a part of the power is returned to the hydraulic system through the sun gear $s_1$ to form cycle power, and a part of the power is transmitted to the rear row through the planet carrier $c_1$ and finally output by the ring $r_3$ of the $P_3$ planetary row. When there is no cycle power in the closed hydraulic system, the power transmission route is as shown in Figure 8b. At this time, the mechanical power is input by the ring $r_1$ of the $P_1$ planetary row, and the hydraulic power is input by the sun wheel $s_1$; after confluence, the power is transmitted to the rear row through the planet carrier $c_1$ and finally output by the ring $r_3$ of the $P_3$ planetary row.

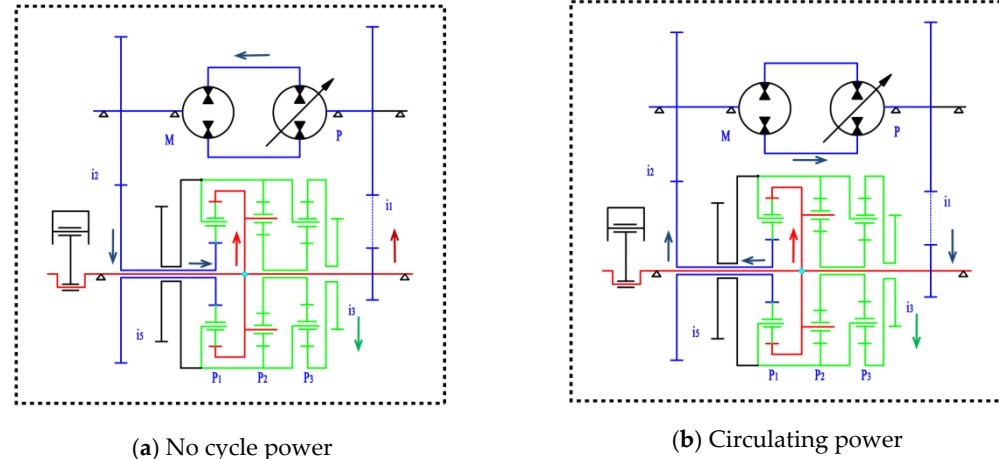

(**a**) No cycle power     (**b**) Circulating power

**Figure 8.** Power flow characteristic diagram of HM1 section.

The HMCVT realizes the continuous change in transmission ratio through the confluence of hydraulic power and mechanical power. The efficiency of hydraulic transmission is far lower than that of mechanical transmission, so the proportion of hydraulic transmission power directly affects the transmission efficiency of the HMCVT system [34]. The power output ratio $\rho$ of the hydraulic circuit is defined as the ratio of the power transmitted to the planetary gear mechanism through the hydraulic circuit to the output power of the HMCVT system. Taking the planetary row as the node analysis, the input power to the planetary row is defined as positive, and the output power of the planetary row is negative. The calculation of $\rho$ is shown in Formula (15).

$$\rho = -\frac{P_3}{P_2} \tag{15}$$

where $P_3$ is the power transmitted through the hydraulic circuit to the planetary gear mechanism and $P_2$ is the output power of the HMCVT system.

For the scheme in this paper, when $\rho = 0$, the transmission is purely mechanical, and the hydraulic circuit pump–motor system is not involved in power transmission. When $\rho \neq 0$, the power flow of the HMCVT system is simplified to two situations, as shown in Figure 9.

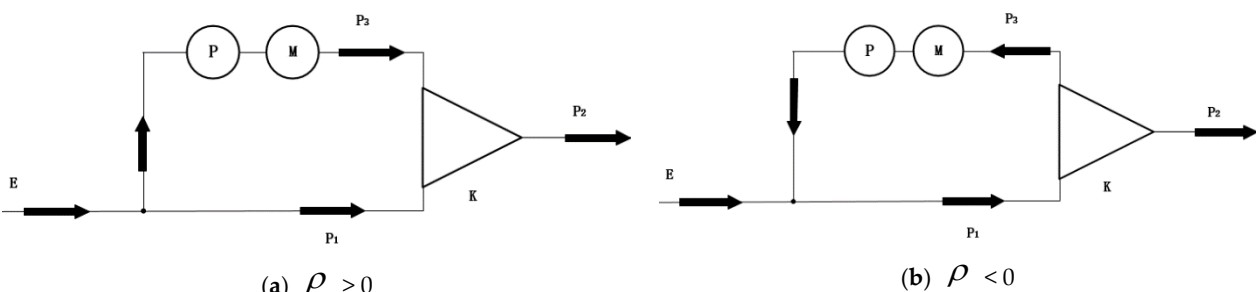

(**a**) $\rho > 0$     (**b**) $\rho < 0$

**Figure 9.** HMCVT power split diagram.

When $\rho > 0$, the system is in the state of hydraulic mechanical power shunt. After the power is output by the engine, part of the power is input to the planetary row after the torque is divided by the hydraulic circuit, and part of the power is directly input to the planetary row by the mechanical circuit. The two power sources flow through the three planetary rows and output the system. When $\rho < 0$, the system is undergoing the hydraulic power cycle. After the power is outputted by the engine, it is directly input into the planetary row through the mechanical circuit; a part of the power is outputted by the

planetary row system, and a part of the power is input into the hydraulic circuit by the planetary row and returned to the power input shaft to form the hydraulic cycle power.

The torque of each component of the planetary gear mechanism has the vector relationship shown in Formula (16).

$$T_s : T_r : T_c = 1 : k : -(1+k) \tag{16}$$

where $T_s$ is the sun wheel torque, $T_r$ is the ring torque, and $T_c$ is the planet carrier torque.

For this design scheme, there are two groups of linkage parts, $c_1$–$r_2$–$c_3$ and $s_2$–$s_3$, which have the following torque relationships:

$$T_{c1} + T_{r2} + T_{c3} = 0 \tag{17}$$

$$T_{s2} + T_{s3} = 0 \tag{18}$$

where $T_{c1}$ is the $P_1$ planet carrier torque, $T_{s2}$ is the $P_2$ sun wheel torque, $T_{r2}$ is the $P_2$ ring torque, $T_{s3}$ is the $P_3$ sun wheel torque, and $T_{c3}$ is the $P_3$ planet carrier torque.

By definition, $\rho$ is the ratio of the input power from the $P_1$ sun wheel $s_1$ to the output power of the HMCVT system. The torque $T_{s1}$ of the $P_1$ sun wheel $s_1$ at each working section can be derived from Formulas (16)–(18), and the speed of the $P_1$ sun wheel $s_1$ is easily known as $n_{s1} = \frac{n_e e}{i_1 i_2}$ from the transmission scheme.

Substituting the input power of the $P_1$ sun wheel $s_1$ and the output power of the HMCVT system into Formula (15), the power output of three hydraulic circuits is calculated as shown in Formulas (19)–(21).

$$\rho_{HM1} = \frac{(1 + k_2 + k_3)\frac{e}{i_1 i_2}}{(1 + k_2 + k_3)(k_1 + \frac{e}{i_1 i_2}) - (1 + k_1)(1 + k_2)} \tag{19}$$

$$\rho_{HM2} = \frac{k_2 \frac{e}{i_1 i_2}}{(1 + k_1)(1 + k_2) - k_2(k_1 + \frac{e}{i_1 i_2})} \tag{20}$$

$$\rho_{HM3} = \frac{\frac{e}{i_1 i_2}}{k_1 + \frac{e}{i_1 i_2}} \tag{21}$$

According to Formulas (19)–(21), combined with the transmission parameters in Table 2, the power output ratio characteristics of the hydraulic circuit are obtained, as shown in Figures 10 and 11.

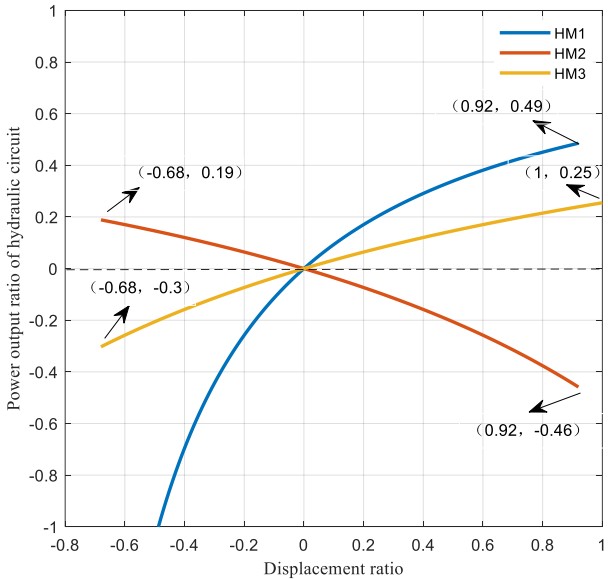

**Figure 10.** Relationship between power output ratio and displacement ratio e of the hydraulic circuit.

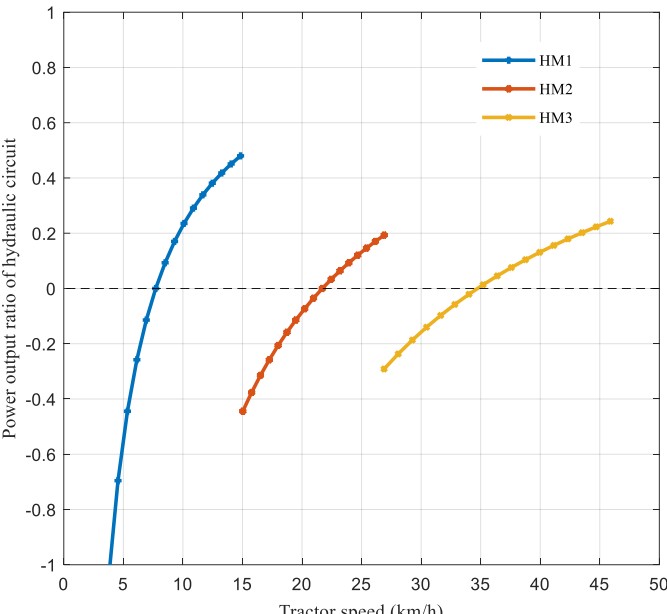

**Figure 11.** Relationship between power output ratio of the hydraulic circuit and vehicle speed.

In Figure 10, the three sections of the hydraulic circuit power output ratio show that there are three points of 0, the hydraulic circuit pump–motor system does not participate in power transmission, the system is pure mechanical transmission, and the transmission efficiency is the highest. HM1 is the starting stage. In order to obtain a high transmission ratio and realize low-speed and high-torque operation, the cycle power of the system is higher. With the decrease in displacement ratio, the transmission power ratio of the hydraulic circuit gradually decreases. When the hydraulic circuit power output ratio $\rho > 0$, there is no cycle power in the system. The highest power output ratios of the three hydraulic circuits are 0.49, 0.19, and 0.25, respectively. When $\rho < 0$, the cycle power of the system is expressed. The highest cycle power ratios of the hydraulic circuit in the HM2 and HM3 sections are 0.46 and 0.3, respectively, which are lower than 0.5.

It can be seen from Figure 11 that there are working intervals with and without the cycle power in each working section. The cycle power, as a useless power with no output system, affects the overall transmission efficiency of the system; it is necessary to obtain as high a work efficiency as possible to ensure that the tractor works in the no-cycle-power range.

*3.4. Results of Efficiency Characteristic*

The HMCVT is a closed planetary gear transmission. When the power flow direction changes, the system will generate cycle power, which does not output the system and affects the overall transmission efficiency [35]. Many factors affect the transmission efficiency of the HMCVT, such as gear friction loss, pump–motor system loss, oil lubrication loss, and heat loss. This paper mainly considers the loss of two high power flows, namely pump–motor system loss and gear friction loss (fixed-axis gear and planetary gears) [36,37]. According to conservation, the power loss of the HMCVT system satisfies the relationship shown in Formula (22).

$$\Delta P = \Delta P_H + \Delta P_J = -P_2\left(\frac{1}{\eta_{HM}} - 1\right) \tag{22}$$

where $\Delta P_H$ is the power loss of the pump–motor hydraulic circuit, $\Delta P_J$ is the power loss of the planetary gear mechanism, and $\eta_{HM}$ is the transmission efficiency of the HMCVT system.

When the system does not have cycle power, $0 < \rho < 1$, the power flow is as shown in Figure 8a and the system power loss is as follows:

$$\Delta P_H = \frac{P_3}{\eta_H} - P_3 = P_3(\frac{1}{\eta_H} - 1) = -P_2\rho(\frac{1}{\eta_H} - 1) \tag{23}$$

$$\Delta P_J = -\frac{P_2}{\eta_J} - (-P_2) = -P_2(\frac{1}{\eta_J} - 1) \tag{24}$$

Substitute Formula (22):

$$\eta_{HM} = \frac{\eta_H \eta_J}{\rho(\eta_J - \eta_H \eta_J) + \eta_H} \tag{25}$$

When the system has cycle power, $\rho < 0$, the power flow is as shown in Figure 8b and the power loss of the system is as follows:

$$\Delta P_H = P_3 = P_2\rho \tag{26}$$

$$\Delta P_J = -P_2(\frac{1}{\eta_J} - 1) \tag{27}$$

Substitute Formula (22):

$$\eta_{HM} = \frac{\eta_J}{1 - \eta_J\rho} \tag{28}$$

In the power transmission process of the hydraulic circuit, power loss occurs through $i_1$ and $i_2$, two pairs of fixed-axis gears, and power loss occurs through $i_3/i_4/i_5$ and $i_6$, two pairs of fixed-axis gears, after confluence. Upon combining the power output ratio of the hydraulic circuit (19)–(21) and the efficiency calculation Formulas (25) and (28), the transmission efficiency expressions of the three HMCVT sections are obtained as follows:

$$\eta_{HM1} = \begin{cases} \eta_{i3}\eta_{i6}\frac{\eta_{J1}}{1-\rho}; -1 < e < 0 \\ \eta_{i3}\eta_{i6}\frac{\eta_{i1}\eta_{i2}\eta_H\eta_{J1}}{\rho(\eta_{J1}-\eta_{i1}\eta_{i2}\eta_H\eta_{J1})+\eta_{i1}\eta_{i2}\eta_H}; 0 < e < 1 \end{cases} \tag{29}$$

$$\eta_{HM2} = \begin{cases} \eta_{i4}\eta_{i6}\frac{\eta_{i1}\eta_{i2}\eta_H\eta_{J2}}{\rho(\eta_{J2}-\eta_{i1}\eta_{i2}\eta_H\eta_{J2})+\eta_{i1}\eta_{i2}\eta_H}; -1 < e < 0 \\ \eta_{i4}\eta_{i6}\frac{\eta_{J2}}{1-\rho}; 0 < e < 1 \end{cases} \tag{30}$$

$$\eta_{HM3} = \begin{cases} \eta_{i5}\eta_{i6}\frac{\eta_{J3}}{1-\rho}; -1 < e < 0 \\ \eta_{i5}\eta_{i6}\frac{\eta_{i1}\eta_{i2}\eta_H\eta_{J3}}{\rho(\eta_{J3}-\eta_{i1}\eta_{i2}\eta_H\eta_{J3})+\eta_{i1}\eta_{i2}\eta_H}; 0 < e < 1 \end{cases} \tag{31}$$

where $\eta_H$ is the pump–motor transmission efficiency, $\eta_J$ is the confluence mechanism transmission efficiency, and $\eta_i$ is the fixed-axis-gear transmission efficiency.

The HMCVT structure is complex and consists of a high number of components, requiring high manufacturing and assembly accuracy [38]. Therefore, the fixed shaft gear is designed according to the seven-level accuracy requirement, so $\eta_i = 0.98$. The transmission efficiency of a single planetary gear can be calculated by the conversion gear train method as 0.96. In the HM1 section, three planetary gear mechanisms participate in the power transmission, so $\eta_{J1} = 0.96^3$. In the HM2 section, two planetary gear mechanisms participate in the power transmission, so $\eta_{J2} = 0.96^2$. In the HM3 section, only one planetary gear mechanism participates in the power transmission, so $\eta_{J3} = 0.96$. The relationship between the transmission efficiency and the displacement ratio of the three-planetary-row linkage HMCVT is obtained using Formulas (29)–(31), as shown in Figure 12.

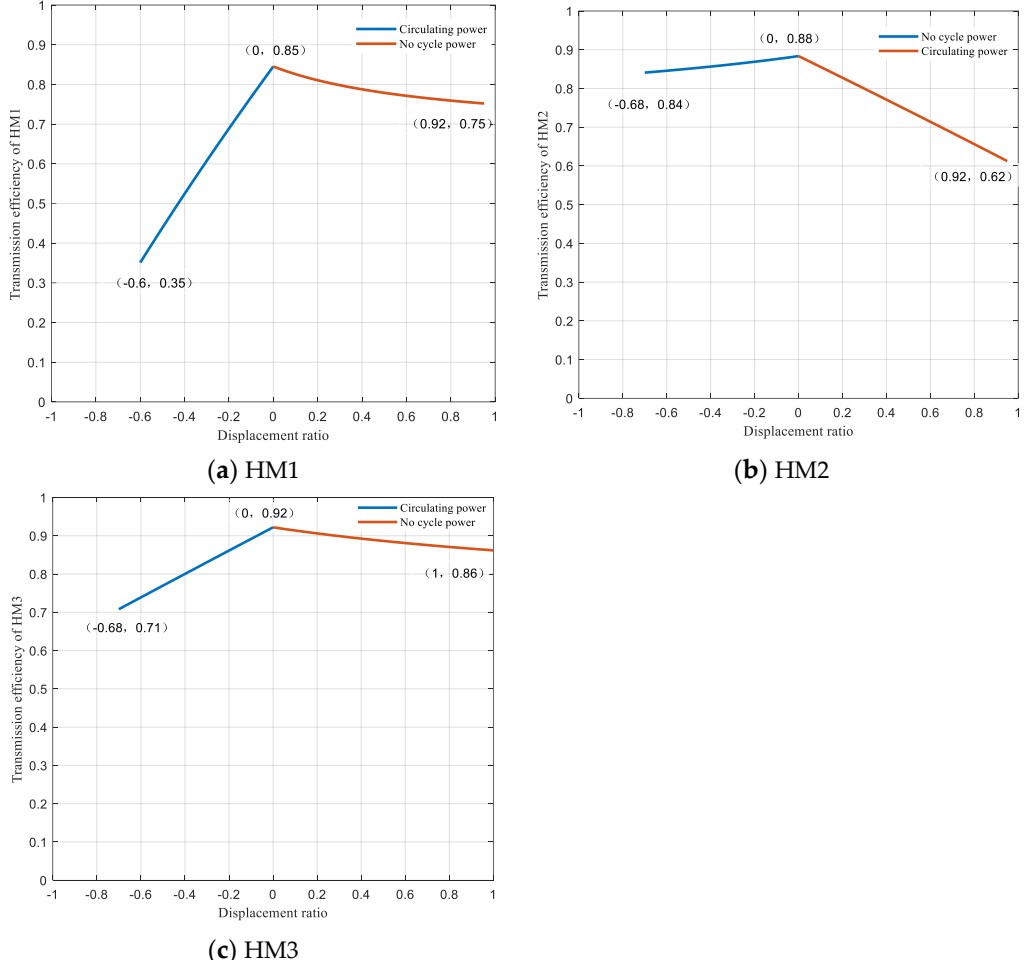

**Figure 12.** HMCVT transmission efficiency characteristic diagram.

The overall transmission efficiency of the three sections gradually increases. The highest transmission efficiencies of the three sections are as follows: HM1, 0.85; HM2, 0.88; HM3, 0.92. When there is no cycle power in the HMCVT system, the transmission efficiency decreases with the increase in displacement ratio, which is mainly due to the increase in the displacement ratio and the increase in the proportion of the low-efficiency hydraulic circuit participating in the power transmission, resulting in a decrease in the overall transmission efficiency of the HMCVT. When the HMCVT system has cycle power, its transmission efficiency shows a more obvious decreasing trend with the increase in displacement ratio. At this time, this is mainly due to the generation of cycle power that does not output the system, resulting in a significant decrease in the overall transmission efficiency. Therefore, in the actual operation process, the system can work as far as possible in the no-cycle-power section by adjusting the displacement ratio.

## 4. Conclusions

Looking at the demands for tractors with low-speed and high-torque operation, a three-planetary-row linkage HMCVT scheme was designed. Based on the principle of the minimum output power ratio of the hydraulic circuit, the hydraulic circuit power is transmitted to the sun wheel $s_1$ of $P_1$, the power transmission ratio of the hydraulic circuit is the smallest, and the overall transmission efficiency is the highest.

The CV and CR clutches, which control the forward and backward movements, are placed on the power output shaft. The speeds of the forward and backward sections are symmetrical, with a wide speed range, so that the tractor can operate under complex conditions.

Based on the idea of "type-equivalence" design, the HM1 working section is designed as a hydraulic mechanical power, which can output a wide transmission ratio at a low starting speed. When the displacement ratio $e = -0.9$, the maximum transmission ratio can reach 46 and the maximum torque can reach 4000 N·m, which can meet tractors' low-speed and high-torque operation requirements.

The overall transmission efficiency of the HM1 section is the lowest, and that of the HM3 section is the highest. When there is cycle power, the transmission efficiency of the system significantly decreases with the increase in displacement ratio. When there is no cycle power, the transmission efficiency of the system is less affected by the displacement ratio. The highest transmission efficiencies of the three sections are as follows: HM1, 0.85; HM2, 0.88; HM3, 0.92, with a high overall transmission efficiency.

## 5. Patents

Lu Zhixiong, Zhou Huadong, Zhang Haijun, Qian Jin, Xiao Maohua, Lu Kai, Chen Yuan, Sun Xiaoxu, Qian Yu. A three-planetary three-stage hydraulic mechanical stepless transmission of a high-power tractor. Jiangsu province: ZL202111306679.3, 2022-06-03.

**Author Contributions:** Conceptualization, H.Z. (Huadong Zhou), J.Q. and H.Z. (Haijun Zhang); methodology, H.Z. (Huadong Zhou) and Z.C.; software, H.Z. (Huadong Zhou) and Z.C; validation, H.Z. (Huadong Zhou), Z.C. and Z.L.; formal analysis, H.Z. (Huadong Zhou); investigation, H.Z (Huadong Zhou). and Z.C.; resources, H.Z. (Haijun Zhang) and J.Q.; writing—original draft preparation, H.Z. (Huadong Zhou); writing—review and editing, H.Z. (Huadong Zhou), Z.L. and Z.C.; supervision, Z.L., Z.C. and X.W.; project administration, Z.L. and L.W.; funding acquisition, Z.L. and Y.Z. All authors have read and agreed to the published version of the manuscript.

**Funding:** This research was funded by the Open Project of State Key Laboratory of Tractor Power System in China (SKT2022006) and the National Key Research and Development Plan (2016YFD0701103).

**Institutional Review Board Statement:** Not applicable.

**Informed Consent Statement:** Not applicable.

**Data Availability Statement:** The data presented in this study are available on demand from the corresponding author and first author (at luzx@njau.edu.cn or zhouhuadong1106@126.com).

**Acknowledgments:** The authors thank the National Key Research and Development Plan (2016YFD0701103) and the Open Project of State Key Laboratory of Tractor Power System in China (SKT2022006). We also thank the anonymous reviewers for providing critical comments and suggestions that improved the manuscript.

**Conflicts of Interest:** The authors declare no conflict of interest.

## Nomenclature

| | |
|---|---|
| HMCVT | Hydro-mechanical continuously variable transmission |
| CV, CR, $C_1$~$C_4$ | Clutch in HMCVT system |
| $i_1$~$i_8$ | Gear ratio in HMCVT system |
| $P_1$~$P_3$ | Planetary gears in HMCVT system |
| $k_1$~$k_3$ | Characteristic parameters of planetary gear mechanism |
| HM1~HM3 | Forward power coupled working stages |
| RHM1~RHM3 | Backward power coupled working stages |
| $e$ | Variable pump's displacement ratio |
| $r_d$ | Power radius of the tractor drive wheel |
| $i_q$ | Rear axle transmission ratio |
| $i_b$ | Wheel drive ratio |
| $s_1$~$s_3$ | Sun wheel |
| $r_1$~$r_3$ | Gear ring |

| | |
|---|---|
| $c_1 \sim c_3$ | Planet carrier |
| $n_e$ | Engine speed |
| $n_s$ | Sun wheel speed |
| $n_r$ | Gear ring speed |
| $n_c$ | Planet carrier speed |
| $T_s$ | Sun wheel torque |
| $T_r$ | Gear ring torque |
| $T_c$ | Planet carrier torque |
| $\alpha$ | Engine throttle opening |
| $\rho$ | Power output ratio of hydraulic circuit |
| $P_1$ | Output power of engine |
| $P_2$ | Output power of HMCVT system |
| $P_3$ | Power transmitted through the hydraulic circuit |
| $\Delta P_H$ | Power loss of pump–motor hydraulic circuit |
| $\Delta P_J$ | Power loss of planetary gear mechanism |
| $\eta_{HM}$ | Transmission efficiency of HMCVT system |
| $\eta_M$ | Pump–motor transmission efficiency |
| $\eta_J$ | Transmission efficiency of the confluence mechanism |
| $\eta_i$ | Fixed-axis-gear transmission efficiency |

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
