# Peer review of "Transmission Parameter Design and Characteristic Analysis of Three-Row Parallel Planetary Gear HMCVT"

_machines, doi:10.3390/machines10090740_

Round 1

Reviewer 1 Report

A three-planetary row linkage HMCVT transmission scheme is designed for the operation requirements of low speed and high torque of tractors. The HM1 section adopts hydraulic mechanical power to start operation, so that the tractor can obtain greater transmission ratio at low speed. The HM2 and HM3 sections adopt “equal ratio” design, so that the system has better speed regulation performance. The clutchs controlling forward and backward are placed on the output shaft, so that the forward and backward sections have a wider speed range. The studies carried out are reliable and worthy of approval. New positive results have been obtained: the range of speed control and overcome resistance moments has been expanded. The article is recommended for publication.

At the same time, it should be noted that the theory of mechanisms and machines continues to develop and creates qualitative changes that lead to the achievement of new effects.

Currently, two approaches are used to create CVTs:

1) Using a stepped gearbox in combination with a hydrodynamic converter.

2) Use of stepless gear mechanism.

The first approach is based on the use of a scheme with one degree of freedom at each stage and automatic speed control by a hydrodynamic converter in a limited range of each stage. This approach requires a gearshift mechanism and a hydraulic control system.

The second approach is based on the use of a stepless mechanical system with two degrees of freedom. The presence of two degrees of freedom theoretically provides the entire required range of regulation without the use of a control system. Numerous attempts by inventors to create a CVT based on a two-moving system ended in failure, since the two-moving system does not have motion definability.

The latest theoretical studies of the author have solved the problem of motion definability [Ivanov K.S. Prospects of Creation of Mechanisms with Two Degrees of Freedom. IFToMM World Congress on Mechanism and Machine Science. Advances in Mechanism and Machine Science. Springer Nature Switzerland AG 2019. https://doi.org/10.1007/978-3-030-20131-9_93. PP 937-946]. A new constraint was found that imposes an additional constraint but retains the number of degrees of freedom. This connection combines the force and speed of the contacting links, for example, in a friction joint. Possible options for force-speed communication have been developed and patented.

The use of an additional force-speed connection allows you to create simple and reliable adaptive mechanical systems with the property of self-regulation.

It can be assumed that in the future the use of definable mechanical systems with two degrees of freedom will lead to the achievement of fundamentally new useful results in the field of creating adaptive machine drives.

Author Response

Thank you for your suggestions and opinions. I think it will be very helpful for my next research work.

Reviewer 2 Report

Dear Editor,

The paper entitled 'Transmission Parameter Design and Characteristic Analysis of Three-Row Parallel Planetary Gear HMCVT' has been reviewed and you can find my comments as follows:

The paper is mainly about the CVT system design for tractors as heavy vehicles. In general, the manuscript's topic has a critical issue for minimum energy consumption and the paper is well written and organized. The mathematical model of the dynamic system is obtained correctly and the results are presented finely. I can propose this study as a research paper for your journal after the following minor revisions:

-In Table 2, why is the transmission rate selected as constant? Normally, these ratios are determined for the optimum operation line of the motor considering minimum energy consumption. This point must be clarified. 

-It should be clearly stated by the author the difference between the work [1].

-It is expected to compare the result with the present study. Thus, the proposed system should be compared to show its effectiveness. The following paper can be helpful [2].

-Is there any possibility to add an optimization algorithm to define the design and control parameters?

-Some statements seem long and the English of the paper should be checked again for better reading. Also, the introduction part can be improved as well.

[1]A new non-geometric transmission parameter optimization design method for HMCVT based on improved GA and maximum transmission efficiency. Comput. Electron. Agric. 2019

[2] A comparative study of energy consumption and recovery of autonomous fuel-cell hydrogen–electric vehicles using different powertrains based on regenerative braking and electronic stability control system, MDPI Applied Sciences, 2021

Reviewer 3 Report

-Terminology inaccuracies should be corrected (e.g., it should be Ring gear instead of Gear ring).

-Grammar and spelling/typographical errors should be corrected (e.g., it should be clutches instead of clutchs, on pages 1, 2, 7).

-Some of the symbols within the text are not aligned properly (see the text).

Reviewer 4 Report

HMCVT should be explained once before using the abbreviation.

The article needs a thorough language check. 

Figure 2 should not break over two pages if possible. The complete comment, especially the power routing color coding should be placed before the figure.

In chapter 2.2. the rear axle differential ratio is said to be 0,38 which means that there is a roughly 1 to 2.5 multiplication on the rear axle followed by a 7 to 1 reduction in the hub boxes. Is that correct?

The displacement ratio e in equations 2-7 is introduced abruptly without explanation after the equations.

The "equal ratio" principle needs explanation.

Equation 14 should use the pi symbol.
